# Protocol for developing core outcome sets for evaluation of psychosocial interventions for children and families with experience or at risk of child maltreatment or domestic abuse

Emma Howarth,[1,2] Claire Powell [ORCID],[2] Jenny Woodman [ORCID],[3] Erin Walker,[4] Heather Chesters,[2] Eszter Szilassy,[5] Ruth Gilbert,[6] Gene Feder [ORCID] [7]

For numbered affiliations see end of article.

**Correspondence to**
Dr Claire Powell;
c.powell@ucl.ac.uk

## ABSTRACT

**Introduction** Recognition that child maltreatment (CM) and domestic violence and abuse (DVA) are common and have serious and long-term adverse health consequences has resulted in policies and programmes to ensure that services respond to and safeguard children and their families. However, high-quality evidence about how services can effectively intervene is scant. The value of the current evidence base is limited partly because of the variety of outcomes and measures used in evaluative studies. One way of addressing this limitation is to develop a core outcome set (COS) which is measured and reported as a minimum standard in the context of trials and other types of evaluative research. The study described in this protocol aims to develop two discrete COSs for use in future evaluation of psychosocial interventions aimed at improving outcomes for children and families at risk or with experience of (1) CM or (2) DVA.

**Methods and analysis** A two-phase mixed methods design: (1) rapid reviews of evidence, stakeholder workshops and semistructured interviews with adult survivors of CM/DVA and parents of children who have experienced CM/DVA and (2) a three panel adapted E-Delphi Study and consensus meeting. This study protocol adheres to reporting guidance for COS protocols and has been registered on the Core Outcome Measures for Effectiveness Trials (COMET) database.

**Ethics and dissemination** We will disseminate our findings through peer-reviewed and open access publications, the COMET website and presentations at international conferences. We will engage with research networks, journal editors and funding agencies to promote awareness of the CM-COS and DVA-COS. We will work with advisory and survivor and public involvement groups to coproduce a range of survivor, policy and practice facing outputs.

Approval for this study has been granted by the Research Ethics Committee at University College London.

## INTRODUCTION

Widespread recognition that child maltreatment (CM) and domestic violence and abuse (DVA) are common and have serious and

### Strengths and limitations of this study

► To our knowledge this is the first attempt to develop core outcome sets to address family violence and abuse.
► The study draws on diverse evidence sources and includes people with lived experience, practitioners and policy-makers, as well as researchers.
► This study provides the opportunity to consider the overlap in outcomes sought across two different but related exposures.
► This study is limited by the lack of direct involvement of children and young people.
► It is beyond the means of the study to involve survivors and service providers from low-income and middle-income countries (LMICs), although we will include research from LMICs in the evidence reviews and actively recruit researchers from or researching LMIC settings.

long-term adverse health consequences[1 2] has resulted in policies and programmes to ensure that services respond to and safeguard children (and their families) at high risk of or with experience of CM and/or DVA.[3–6] However, high-quality evidence about how services can effectively intervene is scant.[7–9]

The value of the current evidence base is limited partly because of the variety of outcomes and measures used in evaluative studies.[7 8] This hampers the ability to aggregate evidence pertaining to one particular type of intervention, so as to build a comprehensive picture of its effectiveness when delivered to different populations or in different contexts. Similarly, it is challenging to make comparisons between different types of interventions, which purport to address the same problem within the same group of individuals.[10 11]

More fundamentally, outcomes measured in CM and DVA intervention studies are often a poor or partial reflection of the concepts of success held by those who use, deliver and pay for interventions.[7 8 12] The ultimate goal of intervention studies is to identify interventions that can benefit individuals, families and communities in the future. Therefore, it is crucial that they measure outcomes reflecting the priorities and expectations of these groups so the evidence they generate is relevant to consumers. Outcomes also need to resonate with the priorities of policy-makers and service providers, else effective interventions may be overlooked by those responsible for funding and/or delivery decisions and never commissioned or implemented.[13 14]

Together, these issues mean it is difficult to extract the information needed to inform real-world decisions about which CM/DVA interventions to commission and scale and which to stop funding.

One way of addressing the limitations set out above is to develop a core outcome set (COS), a standardised set of outcomes that researchers, providers, service users and commissioners consider critical or important outcomes in the management of a condition or in this case, a complex public health challenge.[11 15] The COS is then measured and reported, as a minimum standard in the context of trials or other types of research and evaluation[15] and sometimes practice-based monitoring.[16] The aim is to enhance the methodological standard and utility of research in the field, by increasing consistency and reducing reporting bias (where many outcomes are measured and only favourable effects reported) and ensuring the views of important constituencies influence the selection of outcomes to be included in the COS.[10]

The idea of the COS as a mechanism for improving evidence quality has gathered momentum over the past decade since the establishment of the Core Outcome Measures for Effectiveness Trials (COMET) initiative in 2010 (www.comet-initiative.org).[15] While the number of COSs being developed has increased steadily,[16 17] it is clear that studies have mostly focused on COS development for specific health conditions, pharmacological or surgical interventions and/or discrete interventions delivered in healthcare settings.[16 17] In contrast, there has been relatively less focus on the development of COSs in relation to public health problems that require complex multisectoral responses, often delivered to whole families or multiple members of the same family.

### Current study

The study sets out to develop two discrete COSs for use in future evaluation of psychosocial interventions, which aim to improve outcomes for children and families at risk of or with experience of CM or DVA. We use the term 'at risk' so as not to limit the scope of this work to those interventions delivered to families following substantiated experience of CM or DVA or where children and families define their experiences as such but to include interventions offered to families where it is suspected that

an exposure may have taken place or where children's experiences are thought to be on a trajectory towards this.

Children's experiences of CM and DVA frequently overlap[18] and experience of DVA is often conceived of as a type of maltreatment in its own right or a feature of emotional maltreatment.[19 20] Nevertheless, the conceptualisation and response to these two types of trauma can be different, despite similar consequences. For example, there is variation as to whether exposure to DVA is considered as a form of CM. Where DVA is considered as a form of CM, evidence suggests there may be different levels of state intervention where the primary concern is exposure to DVA versus experience of CM.[19 20] This provides the rationale for developing separate outcome sets, however we will explore where the derived outcome sets overlap with a view to identifying outcomes that can be measured in family contexts where both CM and DVA occur. This is a move away from a focus on single-problem areas towards recognition of the constellation of risks often experienced by children and their families.

## METHODS AND ANALYSIS

This study protocol adheres to reporting guidance for COS protocols[21] and has been registered on the COMET database.

### Scope of outcome sets

The CM-COS and the DVA-COS will be developed to support evaluation of the impact of targeted child and/or family focused psychosocial interventions or services, in the context of both research (randomised and non-randomised studies) and practice (service evaluations and monitoring).

The target population for interventions is children aged less than 19 years of age with experience of (current or previous) DVA or CM. Given that many interventions aiming to improve child outcomes do so via support delivered to parents or multiple family members (rather than directly to the child),[7 8 22] the target group also includes parents or families of children experiencing CM or DVA.

We use a definition of psychosocial interventions set out by the Institute of Medicine.[23]

Interventions within the scope of this study include psychotherapies (eg, cognitive-behavioural therapy), community-based treatments, family/systemic therapy, vocational rehabilitation, peer support services, integrated care interventions and out-of-home care (ie, foster care or adoption). Interventions may be delivered in one or more contexts (eg, clinic, school, community). Interventions may be individual, dyad or group based or a combination and delivered to children with or without their parents, to parents alone, to family groups or some combination. To be in scope, an intervention must implicitly or explicitly aim to improve child outcomes by one or more of the following mechanisms: (1) reducing the risk of CM/DVA occurring/reoccurring in the family; (2) improving parental (non-harming and/or harming) functioning

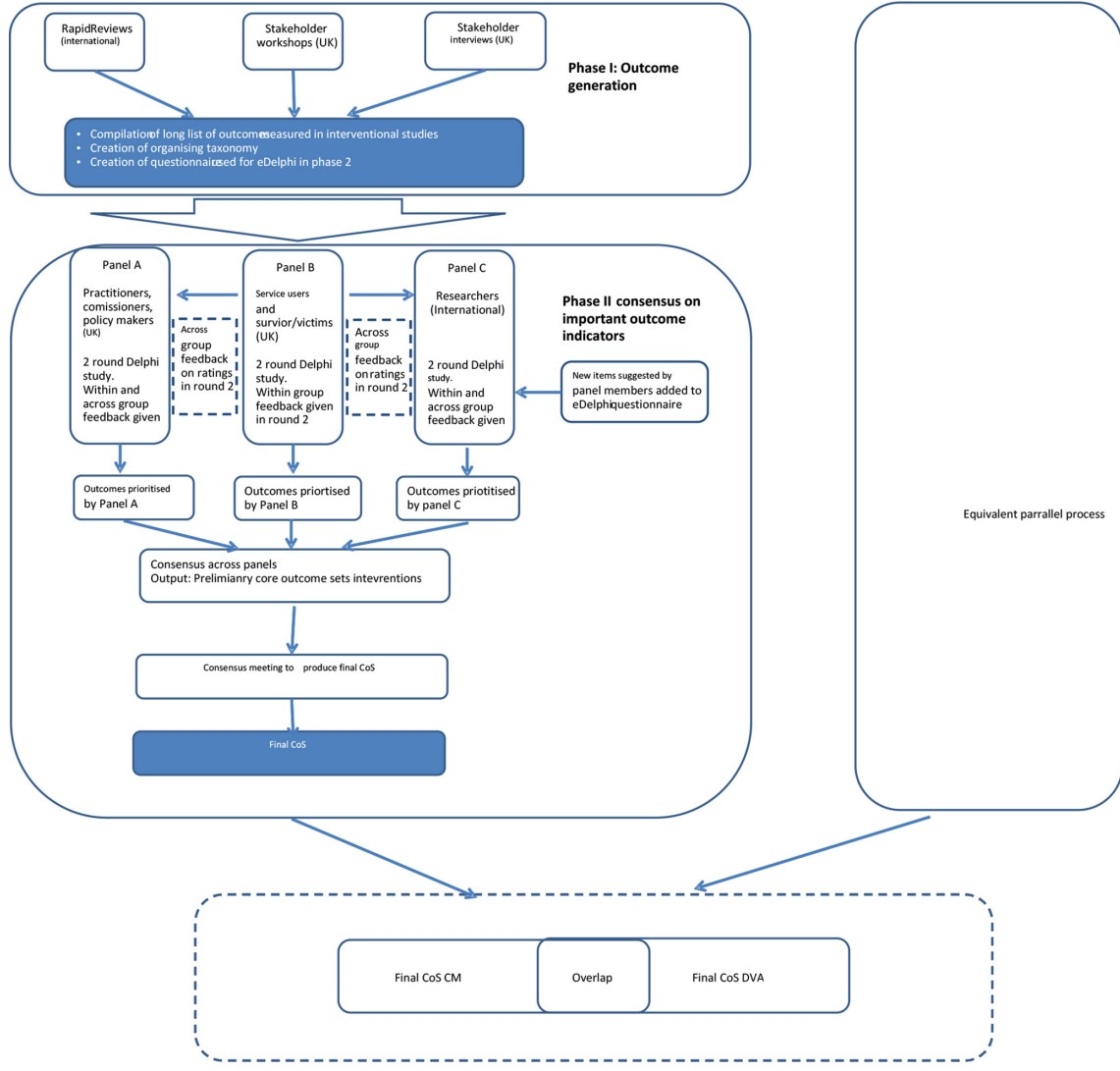

**Figure 1** Study design. CM, child maltreatment; COS, core outcome set; DVA, domestic violence and abuse.

as an indirect route to improving child outcomes; (3) limiting or preventing poor mental health, reduced well-being or function in children following exposure and (4) promoting children's recovery following experience of CM or DVA—here, we relate to the recovery model definition which emphasises perceptions of resilience, self-identity, a sense of empowerment, hope and optimism (eg, Leamy *et al*[24]). Universal and targeted structural interventions are not in scope.

### Study design

The study is being undertaken in two stages (see figure 1). The first stage is underway and seeks to identify candidate outcome areas, domains and indicators. Multiple methods are being used to identify items for the candidate list including rapid evidence reviews, consultation with key stakeholders and qualitative interviews. Data will be synthesised to produce a taxonomy of outcomes, from which the two candidate lists of indicators (structured by area and domain) will be produced.

The second stage, which began in April 2021, will incorporate an adapted two-round E-Delphi Study and

consensus meeting, with the aim of building agreement between different stakeholder groups regarding important outcomes. The E-Delphi technique is an iterative, multistage, online process designed to seek opinion from and develop consensus among a defined group of individuals (panel). The method is frequently used when evidence in an area is known to be limited or contradictory and is widely used in health and social care research. Key features include (1) an anonymous survey process, whereby a panel (or multiple panels) of experts (by profession and/or experience) use a questionnaire to rate a series of statements over a number of rounds; (2) the provision of structured feedback to panel members between rounds with the ability to adjust ratings in light of knowledge about the group opinion and (3) anonymity for panel members during the process.[25] These features can facilitate the convergence of opinion across rounds, helping to build consensus while at the same time highlighting areas of continuing disagreement. This method has been used extensively in the context of core outcomes research.[16 26 27]

We will recruit three panels for participation in the E-Delphi Study to ensure that each stakeholder group is equally represented in the final consensus.[28] In a further effort to ensure that the views of those with lived experience remain a central focus during this exercise, the E-Delphi method will be adapted so that in addition to feedback about their individual and own panel scores for each item, professional and researcher panels will also receive feedback about the scores of the lived experience panel. This adaptation is informed by evidence that feedback of patient scores to clinicians results in an expanded set of consensus items that better reflect the priorities of patients.[29] Additional feedback will not be given to the lived experience panel, so as to minimise the possibility of perceived power differentials influencing this panel's ratings.[28] A final face-to-face consensus meeting will be used to review and verify findings from the E-Delphi Study, clarify any remaining uncertainty and ratify the final COS.

### Study oversight

A steering committee including practitioners, policy-makers and researchers representing CM and DVA fields has been formed and will meet formally two times per year.

### Patient and public involvement

Three public advisory groups are also overseeing and consulting on the study. One group is comprised of individuals with lived experience of DVA and one of care experienced young people. These groups have been formed in partnership with relevant survivor-led organisations. A third group is comprised of young people affiliated to the National Children's Bureau who are consulting more broadly on the work of the Children and Families Policy Research Unit. Partner organisations are funded to organise three meetings per year and to provide appropriate remuneration to participants. Additional funds will be paid to cover scheduled review activities organised with partner agencies via email. Members of advisory groups will be involved in all aspects of the study including the development of the outcomes taxonomy, development of the list of candidate indicators, preparation of materials for the E-Delphi and dissemination of results.

### Participants

Workshops (phase I): We will invite 30–40 individuals to attend each workshop, the aim of which will be to discuss definitions of CM/DVA and outcomes perceived to be important for survivors. Relevant researchers (mainly UK) and professionals from each field (eg, support workers, primary and secondary health practitioners, education staff, local authority commissioners, local and national policy-makers) will be identified from the research team's networks, authorship of key publications and internet searches.

People with lived experience of CM/DVA will be approached via gatekeeper organisations and existing survivor/researcher networks known to the research team. Concerted effort will be made to invite individuals representing groups known to be marginalised from services or research on DVA/CM or who receive inadequate service responses owing to discrimination or lack of service differentiation (ie, assuming all groups require the same response).[30–33]

Semistructured interviews (phase I): We will recruit a sample of approximately five adults who are identified as survivors of CM or exposure to DVA during childhood and five parents of children currently aged 0–18 years with lived experience of DVA/CM. In the first instance, we will seek to recruit participants via gatekeeper organisations (see Procedure section below), although if recruitment is insufficient, we will seek approval for direct recruitment via social and print media. To take part in interviews, participants will be required to self-identify as having experienced CM/DVA or as having a child who has experienced CM/DVA.

Adapted international E-Delphi Study (phase II): Three separate panels will be recruited to take part in the consensus study comprising: (1) individuals with lived experience (parents of children with experience of CM/DVA and adults experiencing abuse in childhood); (2) frontline and strategic professionals involved in the delivery and commissioning of CM/DVA services and related policy and (3) researchers. The first two panels will include members from the UK, with the researcher panel including international researchers from high-income, middle-income and low-income countries. We will aim to recruit 30 individuals to each panel.

Survivors and professionals taking part in the workshops and semistructured interviews described above (and who give consent for further contact) will be approached for participation in the lived experience and professional panels, respectively. If needed, additional participants will be recruited through key organisations working with either CM or DVA survivors and snowball sampling. Key researchers, with at least one peer-reviewed publication from either the CM field or DVA field, will be identified through the rapid reviews, researcher networks, participation in workshops and via the expert panel. For all panels, participants must be able to read and understand English in order to participate.

Consensus workshop following E-Delphi Study (phase II): A face-to-face consensus meeting, with a purposively sampled panel (n=30) representing all key stakeholder groups, will be recruited from participants taking part in earlier phases of the study. Individuals outside of the study will be approached as needed to ensure balanced representation and inclusion of individuals of strategic importance to take up and implementation of study findings. Appropriate amendments to ethical approvals will be sought to accommodate this.

It is important to note that although the focus of this work is on child-targeted and family-targeted interventions, this study does not directly involve children and young people aged <18 years with experience of CM

and/or DVA. We initially explored this possibility with third sector organisations and professionals and clinicians comprising our expert advisory group. However, it was concluded that the nature of this research was not sufficient to justify the potential harm and safeguarding issues that may have been raised by approaching children and young people with recent experience of violence and abuse, particularly as they may not be engaged with supportive services. Instead, the voices of children and young people have been included indirectly via (1) inclusion of outcomes extracted from qualitative studies reporting children and young people's experiences, (2) recruitment of adult survivors of CM and childhood exposure to DVA as well as parents of children with recent experience and (3) consultation with care experienced young people who are advising on the conduct of the study, including review of outcomes identified in the first phase of this work. Nevertheless, the lack of children and young people's direct participation is a limitation to this work, which will be transparently addressed at all stages of reporting.

## Procedure

### Phase I

#### Rapid reviews

We will conduct a series of rapid reviews using systematic methods (see online supplemental appendices for protocols and review questions). We will review experimental and quasi-experimental intervention studies (international), qualitative studies containing primary accounts of experience of relevant interventions or outcomes that are sought by families and children experiencing CM/DVA (international) and the grey (UK) literature reporting descriptions of interventions, service evaluations or consultation regarding appropriate outcomes across the DVA and CM fields.

We will search a range of relevant databases and websites under the guidance of an expert librarian. Following rapid review techniques,[34 35] we will search since 2014 for intervention studies (covering the time elapsed since previous key reviews, Macdonald *et al* and Howarth *et al*[8 36]) and 2005–2014 for the qualitative studies to build on recent qualitative reviews.[37] The grey literature review will primarily focus on searches of relevant UK organisation websites and will include any service or intervention evaluation or any consultation or review, to identify relevant candidate outcomes or outcome tools for use in the context of service delivery or evaluation.

A second reviewer will screen and extract data from a minimum of 5% of titles/abstracts and articles to ensure consistency. Inter-rater reliability kappa scores will be calculated and disagreements will be resolved through discussion (or a third reviewer if necessary) throughout the process. Relevant outcome indicators will be extracted, as well as their measurement instruments where possible. There will be no appraisal of study quality and outcomes will be extracted from all identified papers.

#### Stakeholder workshops

We will hold two invite-only workshops (one focused on CM and one focused on DVA) to gather stakeholder views. The purpose of these events will be to (1) explore definitional issues, specifically how each phenomenon is defined by particular groups and the function that this definition plays in practice (in terms of enabling access to services/interventions and measuring change) and (2) explore outcomes perceived to be important indicators of benefit or harm for children and families experiencing CM/DVA.

Participants will be seated on tables of 6–8. Each table will include at least two individuals with lived experience and one facilitator. Guided by facilitators, participants will be asked to generate ideas relating to desirable (or undesirable) outcomes, unconstrained by what they believe to be measurable or achieved via currently available interventions. This will be an attempt to ensure output is not merely reflective of current practice or discourse. Designated scribes will take notes throughout the day, which will be collated and analysed thematically.[38] Participants in the workshops will be asked for permission to contact them at a later date for the purpose of inviting them to participate in the international E-Delphi Study.

#### Interviews with individuals with lived experience of DVA/CM as a child or as parent of a child

Participants will be identified via key gatekeeper organisations (where work with survivors of CM/DVA is core business) contacted for the purpose of workshop participation (see above). Participants will be approached directly by a professional from the gatekeeper organisation or they will receive an open invitation circulated through the organisation's survivor network. Where participants are approached by professionals, they will be given brief information about the study and asked for permission to pass contact details to the research team. Individuals responding to an open invitation will be asked to contact a member of the research team directly. They will be assured of the anonymity of their involvement.

Basic sociodemographic information and minimal information about experiences of CM or DVA will be collected via questionnaire prior to the interview and will be used for sample description. Participants will have the opportunity to take part in the interview face to face, by video call or by phone, according to their personal preferences and public health guidance on social distancing. For those participants who wish to take part but are unable to speak directly to interviewers, they will be able to answer the interview questions by email.[39] Interview schedules will be used to guide interviews, which will be recorded and transcribed verbatim and analysed thematically.[38]

#### Outcome generation

A list of candidate outcome areas (eg, health and wellbeing), domains (eg, mental health) and specific indicators (eg, withdrawal from friends and activities) will be generated iteratively by the research team, drawing on

all information sources described above. An unedited candidate list of outcome indicators generated from stakeholder workshops will be used as a starting point. Identification of duplicate and overlapping outcome indicators from the list will be undertaken in parallel by two team members (CP, EH). Similar items will be dropped or combined to produce a reduced inventory. Disagreements between team members will be resolved through discussion. All suggestions to drop or combine items will be reviewed by two further research team members (RG, GF) and survivor involvement groups. Similar indicators (ie, outcomes that could be compared across studies or combined in a meta-analysis[21]) will be grouped into outcome domains by two team members and reviewed by two further members of the research team and survivor involvement groups. Simultaneously, a taxonomy to organise domains into broader outcome areas will be developed. Here, we will draw on existing practical and theoretical frameworks to categorise health outcomes,[40] as well as the aetiology and impacts of DVA and CM.[41–44] This overarching framework to describe the hierarchical structure of outcomes identified in workshops will be reviewed and refined by all members of the research team, the expert advisory group and survivor involvement groups.

A candidate list of outcome indicators from the rapid reviews will be generated and deduplicated (CP, EH). Four research team members and at least two survivor representatives will, in parallel, attempt to categorise indicators using the developed taxonomy. Categorisations will be compared, disagreements discussed and consensus reached through discussion. New domains or areas will be added where necessary. Unique indicators (not already included) will be identified from the candidate list generated from the reviews and added to the taxonomy. This iterative process will be repeated with data yielded from interviews.

The final taxonomy and labelling of terms will be reviewed by the advisory group and all three public involvement groups. Particular attention will be given to the language used to describe outcome areas, domains and specific indicators to ensure they are understandable, meaningful and acceptable to all stakeholder groups. Further refinement (including addition of areas, domains or indicators) will be undertaken following this review. The final step in the process will be to examine outcomes against a priori criteria designed to ensure the final COS has maximum utility. These include: (1) the extent to which the outcome indicator relates to children's feelings, function or survival or the process of delivering services to survivors, (2) whether the outcome is 'changeable' and (3) whether the outcome indicator could feasibly change as a result of a psychosocial intervention–here, we will draw on the literature elucidating mechanisms through which exposure to violence and abuse may be communicated to child outcomes (eg, Cameranesi and Piotrowski).[45] Four members of the research team, at least two members of the expert advisory group and four

members of the survivor involvement groups (with equal representation of CM and DVA experience) will independently assess outcome indicators against the criteria listed above. Any indicators identified as not meeting all criteria will be discussed and a decision taken to exclude or include it in the candidate list. Excluded outcomes will be reported in the final paper, along with reasons for exclusion. Where needed, a glossary of terms and explanatory text will be developed to aid clarity for participants in the E-Delphi Study.

### Phase II
#### Adapted international E-Delphi Study
A sequential two-round, three-panel E-Delphi Study will be conducted.

Round 1: A questionnaire for use in the E-Delphi Study will be developed using the taxonomy described above. Areas and domains will serve as headings and subheadings by which to organise the survey, so as to encourage completion and to allow us to explore the relative importance of indicators within the same domain. The questionnaire will be reviewed by advisory and involvement groups and refined in line with feedback. Ethical approval will be sought as an amendment to that granted for phase I of the study.

Participants will be contacted by email to remind them about the COS study and their attendance at a previous workshop (if appropriate) and to invite them to participate in the E-Delphi Study. A second email containing the information sheet and link to an online questionnaire will be sent 1–2 days after the initial contact. Participants will be required to indicate that they have read the information sheet and agree to take part, before proceeding to the questionnaire. The questionnaire will be administered via Qualtrics (https://www.qualtrics.com/uk/) hosted by the University College London.

Participants will be presented with a list of outcome indicators organised by area and outcome domain. They will be asked to rate each outcome presented, on a 9-point scale of importance (1=not at all important, 9=extremely important). Participants will also be given the opportunity to add any additional outcomes that are missing from each domain using a free-text comments box. During this round, we will also collect demographic data including ethnicity, age, gender, profession and country of professional operation. The questionnaire will remain open for 14 days and reminder emails will be sent out at 7 and 2 working days before closure.

Item-level descriptive statistics will be generated for each panel and item including: number of respondents, minimum and maximum values, measures of central tendency and dispersion. Criteria for item inclusion in round two will be an item is rated 7–9 (on a 9-point Likert Scale) by 50% or more participants in at least one panel and 1–3 by no more than 15% of participants in any stakeholder group.[46] This low threshold for inclusion enables us to reduce response burden in round two by dropping unimportant items given higher number of items are

associated with significantly lower response rates in COS Delphi surveys,[47] while also reducing the likelihood of dropping outcomes that may have been rated more highly in subsequent rounds had participants been given feedback on them. New items will be included if two or more panellists suggest inclusion and the research team deem it unique to existing content.[15] Panellists completing round one will be invited to participate in round two if they rated ≥50% of survey items. Non-completers will not be contacted for participation in round two. We will assess attrition rates for each panel and by demographic profiles.

Round two: An amendment to the existing approval will be sought for use of the shorter round two questionnaire. The same items will be included in questionnaires issued to each panel. Each panel member will receive a personalised questionnaire reporting panel averages and their own rating for each item. As noted above, professional and researcher panels will also receive feedback about the ratings of the survivor panel. Panellists will be asked to re-rate each of the included items and rate for the first time any new outcomes put forward in round one. All new outcomes suggested in round one (irrespective of the panel from which they derived) will be presented to each of the three panels.

As before, participants will receive two reminders to complete the questionnaire, over the course of 14 days. Following completion of the study, descriptive statistics will be computed. Items will be deemed important to a particular panel if they are rated 7–9 by ≥70% of respondents and 1–3 ≤15% by the panel. Conversely, items will be classified as unimportant to a group if ≥70% of respondents rate it as 1–3 and ≤15% rate it as 7–9. Any items not classified as important or unimportant will be deemed not to have reached consensus. Items will be considered 'core' and recommended for inclusion in the COS if they are rated as important by all three panels. We will assess the impact of attrition on consensus by comparing (within panels) the mean total item scores for those completing round one only and those completing both rounds; we will also compare the average scores for completers versus non-completers by each item (within panel).[15]

### Consensus meeting
A face-to-face consensus meeting, with a purposively sampled panel (n=30) representing all key stakeholder groups, will be held to discuss, vote and agree on the final CM-COS and DVA-COS. The format of the meeting will follow the process set out by the James Lind Alliance (JLA) final priority setting workshops http://www.jla.nihr.ac.uk/jla-guidebook/chapter-8/workshop-process-on-the-day.htm. This method is pertinent given that JLA priority setting meetings involve multiple stakeholders, discussion of interim results derived from the ranking of evidence uncertainties and production of a 'top ten'.

While there is no recommended maximum number of outcomes that should be included in a COS, for it to be pragmatic we aim to arrive at a maximum of 10

outcomes.[48 49] The JLA priority setting method involves a structured process including small group and whole group discussion, ranking and reranking. The method will be adapted to include a preliminary step, where participants review those outcomes identified as important to the lived experience panel, but which did not reach consensus across all groups. Participants will be asked to identify any outcomes that should be discussed in the workshop, alongside outcomes meeting the consensus definition. This initial step is an attempt to ensure appropriate weight is given to the voice of those with lived experience of DVA/CM. During discussion, workshop participants will be asked to take into consideration the extent to which identified outcomes are 'changeable', and could be feasibly impacted by psychosocial interventions. The final COS and also a list of all items reaching consensus will be published.

## ETHICS AND DISSEMINATION
### Ethical approval
Ethical approval was sought from the Research Ethics Committee at University College London. At all stages of the study, we will obtain written consent for contact information relating to potential participants to be passed via gatekeeper organisations assisting with recruitment. We will obtain written informed consent from participants in interviews and the consensus meeting. Online consent will be obtained from participants when they opt in to participate in the E-Delphi Study, before they are able to proceed.

### Dissemination and implementation
We have registered the study on the COMET website. We will provide tailored briefings to UK policy-makers, think tanks, commissioners and third sector organisations while the study is in progress as well as completed. This will maximise interest and intention to use the COSs. We also intend to use these briefings as a vehicle for recruitment to the E-Delphi Study. We will involve the leads of international scholarly networks in workshops and recruit member networks to the E-Delphi Study.

We will disseminate our findings through peer-reviewed and open access publications, the COMET website and presentations at international conferences. We will engage with journal editors and funding agencies and the relevant Cochrane and Campbell review groups to promote awareness of the CM-COS and DVA-COS. We will provide briefings and links to publications to international research and policy networks, for dissemination through the Violence, Abuse and Mental Health Network (VAMHN) membership and NIHR Children and Families Policy Research Unit (CPRU) collaborators, as well as the wider network of National Institute for Health Research (NIHR) Policy Research Units, Applied Research Collaborations and UK Research and Information (UKRI) networks. We will invite survivors who participated in workshops and in involvement groups to coproduce

plain-language, service-user facing communication materials for circulation in places where survivors access support (formal or informal). We will also develop tailored briefings to enable findings to be shared with all study participants; participation in this type of study is known to be a key facilitator of implementation.[15] Briefings will be published on the CPRU website and emailed to all third sector organisations working specifically with survivors of CM and DVA, as well as local authority commissioners and Clinical Commissioning Groups (CCG).

A high-level review of the reach and uptake of the COSs will be undertaken in 2023. One of the key issues for review will be whether the COS has become aligned or adopted by research and practice networks or collaborations and recognised by funders (eg, NIHR) and bodies coordinating health and social care intervention research and systematic reviews (eg, Cochrane and Campbell Collaborations).

## DISCUSSION

Currently no published COS exists for evaluation of services and interventions to improve child outcomes following experience of CM or DVA. It is essential that outcomes measured in the context of trials and practice-based research reflect the benefits (and harms) sought and prioritised by those who use, deliver and commission DVA and CM programmes, as well as those who research them. A COS that is developed with strong participation from people with lived experience of CM or DVA and those working to support them will help to ensure that relevant outcomes are measured in all evaluative studies. This in turn will enhance consistency across studies and the quality and value of research. High levels of awareness and uptake of this study's outputs are critical to achieve its ultimate aim.

## Limitations

The design of this study is limited by the lack of direct involvement of children and young people in either qualitative interviews or the E-Delphi Study. Given the study described here represents meta-research, it was felt that potential risks to children could not be justified. Their voices are nonetheless to some extent reflected through the broad reviews of evidence and inclusion of parent perspectives. It is also beyond the means of the study to involve survivors and service providers from low-income and middle-income countries (LMICs), although we will include research from LMICs in the evidence reviews and actively recruit researchers from or researching LMIC settings.

## Author affiliations
[1]School of Psychology, University of East London, London, UK
[2]Institute of Child Health, University College London, London, UK
[3]Institute of Education, University College London, London, UK
[4]UCL Partners, University College London, London, UK
[5]Centre for Academic Primary Care, Population Health Sciences, University of Bristol Medical School, Bristol, UK
[6]Centre for Paediatric Epidemiology and Biostatistics, University College London Institute of Child Health, London, UK
[7]Community Based Medicine, University of Bristol Medical School, Bristol, UK

**Acknowledgements** First and foremost the authors would like to thank the survivors and other members of the public who contributed to this study. Survivor involvement is facilitated by VOICES, a survivor-led charity for women who have experienced domestic abuse. The authors also extend their thanks to members of their advisory group who have informed the development of the study design and have commented on drafts of this manuscript. Members of the professional advisory group are: Elaine Fulton, Dr Deborah Hodes, Dr Carol Rivas, Professor Sally Kendall, Professor Geraldine Macdonald, David Carney Haworth, Elisabeth Carney Haworth, Victoria Jepson and Hannah Edwards.

**Contributors** EH conceived of the original study design, which was refined and developed by her, CP, RG, GF, JW, ES and EW. Authors CP and EW led the development of the public patient involvement strategy. CP and HC developed protocols for rapid reviews, which were reviewed and refined by CP, EH, RG, JW and GF. Author CP undertook all searches. CP and EH performed data extraction for reviews. EH, CP, RG, JW, EW, ES and GF contributed to the writing and review of the protocol paper.

**Funding** This study was funded by the National Institute for Health Research (NIHR) Policy Research Programme, funder reference: PR-PRU-1217-21301; UCL award code: 177763. The views expressed are those of the author(s) and not necessarily those of the NIHR or the Department of Health and Social Care.

**Competing interests** None declared.

**Patient consent for publication** Not required.

**Provenance and peer review** Not commissioned; externally peer reviewed.

**ORCID iDs**
Claire Powell http://orcid.org/0000-0002-6581-0165
Jenny Woodman http://orcid.org/0000-0002-9403-4177
Gene Feder http://orcid.org/0000-0002-7890-3926

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
