## [Reviewer comments · BMJ Open]

ARTICLE DETAILS

TITLE (PROVISIONAL)	Protocol for developing Core Outcome Sets for evaluation of psycho-social interventions for children and families with experience or at risk of child maltreatment or domestic abuse
AUTHORS	Howarth, Emma; Powell, Claire; Woodman, Jenny; Walker, Erin; Chesters, Heather; Szilassy, Eszter; Gilbert, Ruth; Feder, Gene

VERSION 1 – REVIEW

REVIEWER	Fattah, Kazi University of Queensland Faculty of Humanities and Social Sciences, Sociology
REVIEW RETURNED	04-Oct-2020

GENERAL COMMENTS	I read the paper with much interest. It presents a clearly articulated and detailed description of the study protocol which I think would be of much use and interest particularly to other researchers using/adopting similar methodologies. I have just a few very minor observations: For the Phase 1 workshops, I would have chosen a smaller number of participants for each workshop (not more than 25 instead of 40) which might be more effective to achieve the desired outputs. Of course, with good facilitation skills and if trained co-facilitators are available then 40 should be manageable. Please watch out for inconsistent/incorrect use of punctuations (period and comma) in the in-text citations. In the Introduction, for example, the citation numbers are sometimes placed before the period (in the same sentence) and other times after the period/comma. Sometimes sentences began with citation numbers in parenthesis which appears very odd. Space is often missing after comma. This can be easily addressed with a round of quick edit. Page 13, lines 272-273: Grammatical error – “Any indicators identified as not meet all criteria by one or more reviewers will be discussed...” In the Procedure section, the issue of maintaining confidentiality was mentioned several times under different subsections. It should be sufficient to mention this in Ethics section only rather than repeating in different sections. I wish the authors success with their endeavours.
--

REVIEWER	Hafstad, Gertrud Norwegian centre for Violence and Traumatic Stress Studies
REVIEW RETURNED	29-Oct-2020

GENERAL COMMENTS	Thank you for the opportunity to review this interesting project. The protocol describes a study set up for developing a core set of outcomes for evaluating the effects of psychosocial intervention for children and families with experiences or at risk for child maltreatment or domestic violence. The protocol is well written, and the rationale for developing a core set of outcomes is well described and founded in the existing literature. The authors present a thorough plan, which will involve a number of experts, stakeholders and individuals with lived experiences. In all, this is a very worthwhile project which is described with a high level of detail. With all the planned effort put into the development of such a core set of outcomes, it is extremely important that the results are disseminated properly so as to increase the likelihood that it is used and thus actually fills the gaps that they are developed to fill. I believe that this may be the most critical point of the planned project. Additionally, the focus on both risk for experiencing and having experienced CM/DV is somewhat broad for a single project like this and may preclude the stringency and usefulness of the outcome sets that are to be developed. As the project seems well planned and the protocol seems thoroughly worked through, I only have a few more specific comments:  1. The point raised in lines 87 and 88 is a little unclear and would probably need a reference- 2.. Page 7, lines 148 onwards, it would be good with some justification of the number of participants to include in the workshop. 30-40 persons seems like a lot of people and pros and cons pertaining to groups size should be discussed briefly with regard to ascertaining that several perspectives are represented but at the same time that all participants will have a realistic chance of being heard. 3. I believe a very brief description of the eDelphi format could be included for readers who are not very familiar with Delphi processes. 4. The authors note that the lack of the direct involvement of children is a limitation. I agree, and I am not sure that strain put in children or adolescents in taking part in a discussion group outweighs the advantage of including the child perspective to increase the validity of the outcome sets developed. 5. As far as I can see, no time plan is presented for the study. This should be included. Minor: Make sure the spelling of COS is consistent (written CoS some places in the text, l. 266) Please check that "prioritised" is the correct word in the sentence on line 380
---

VERSION 1 – AUTHOR RESPONSE

Reviewer: 1

R1 - For the Phase 1 workshops, I would have chosen a smaller number of participants for each workshop (not more than 25 instead of 40) which might be more effective to achieve the desired outputs. Of course, with good facilitation skills and if trained co-facilitators are available then 40 should be manageable.

We have clarified that the format of the workshops is small groups of 6-8.

R1- Please watch out for inconsistent/incorrect use of punctuations (period and comma) in the in-text citations. In the Introduction, for example, the citation numbers are sometimes placed before the period (in the same sentence) and other times after the period/comma. Sometimes sentences began with citation numbers in parenthesis which appears very odd. Space is often missing after comma. This can be easily addressed with a round of quick edit.

We have addressed this inconsistency. In line with journal format, references in parentheses are placed after the full stop.

R1- Page 13, lines 272-273: Grammatical error – “Any indicators identified as not meet all criteria by one or more reviewers will be discussed...”

This has been changed in the text.

R1 - In the Procedure section, the issue of maintaining confidentiality was mentioned several times under different subsections. It should be sufficient to mention this in Ethics section only rather than repeating in different sections.

Mention of confidentiality has been deleted at the bottom of page nine and added to the dissemination and ethics section.

Reviewer: 2

R2- it is extremely important that the results are disseminated properly so as to increase the likelihood that it is used and thus actually fills the gaps that they are developed to fill. I believe that this may be the most critical point of the planned project.

We agree. We have clarified (line 415) that recruitment to the E-delphi is also a mechanism for implementation.

R2- Additionally, the focus on both risk for experiencing and having experienced CM/DV is somewhat broad for a single project like this and may preclude the stringency and usefulness of the outcome sets that are to be developed.

Thank-you for this comment. We have provided further clarification regarding our use of the term ‘at risk’. The focus on a broader set of families, other than those with a substantiated exposure is largely informed by child maltreatment literature and practice where this is a common term used to promote parental engagement. (line 88) “We use the term ‘at risk’ so as not to limit the scope of this work to those interventions delivered to families following substantiated experience of CM or DVA or where children and families define their experiences as such; but to include interventions offered to families where it is suspected that an exposure may have taken place, or where children’s experiences are thought to be on a trajectory towards this.”

R2 1. The point raised in lines 87 and 88 is a little unclear and would probably need a reference-

We have clarified and slightly expanded this point In lines 94-100, adding additional references.

R2 - 2.. Page 7, lines 148 onwards, it would be good with some justification of the number of participants to include in the workshop. 30-40 persons seems like a lot of people and pros and cons pertaining to groups size should be discussed briefly with regard to ascertaining that several perspectives are represented but at the same time that all participants will have a realistic chance of being heard.

We have clarified that the format of the workshops is small groups of 6-8.

R2 3. I believe a very brief description of the eDelphi format could be included for readers who are not very familiar with Delphi processes.

We have added description of the E-delphi method on lines 141-150.

R2 4. The authors note that the lack of the direct involvement of children is a limitation. I agree, and I am not sure that strain put in children or adolescents in taking part in a discussion group outweighs the advantage of including the child perspective to increase the validity of the outcome sets developed.

R2 5. As far as I can see, no time plan is presented for the study. This should be included.

We now provide information about the timing of each of the study phases, as well as when the study is due to report in the abstract and study design section.

R2 - Make sure the spelling of COS is consistent (written CoS some places in the text, l. 266)

This has been corrected.

R2 - Please check that "prioritised" is the correct word in the sentence on line 380

This has been changed to 'sought and prioritised ' (ln 427)

VERSION 2 – REVIEW

REVIEWER	Hafstad, Gertrud Norwegian centre for Violence and Traumatic Stress Studies
REVIEW RETURNED	03-Mar-2021

GENERAL COMMENTS	I think this is an interesting and well-written paper, and I believe the authors have done a good job in revising the manuscript. Although the authors have responded well to almost all my comments, I would like to see a response to the comment concerning the involvement of children (or lack of such) in the process of developing the COS. I still believe this is an important point. We now know the importance of including children's perspectives on issues concerning them, and I think the paper
---

	would need a stronger justification for not consulting young people in the COS development process. That said, I do acknowledge that a discussion about this point has been elaborated in the limitations section, but I still believe that it would strengthen the project if children's perspectives are included.
--	--

VERSION 2 – AUTHOR RESPONSE

Reviewer: 2

Dr. Gertrud Hafstad, Norwegian centre for Violence and Traumatic Stress Studies

Comments to the Author:

I think this is an interesting and well-written paper, and I believe the authors have done a good job in revising the manuscript. Although the authors have responded well to almost all my comments, I would like to see a response to the comment concerning the involvement of children (or lack of such) in the process of developing the COS. I still believe this is an important point. We now know the importance of including children's perspectives on issues concerning them, and I think the paper would need a stronger justification for not consulting young people in the COS development process.

That said, I do acknowledge that a discussion about this point has been elaborated in the limitations section, but I still believe that it would strengthen the project if children's perspectives are included.

Authors' reply:

We thank Dr. Hafstad for persevering with this comment and apologise for not having sufficiently addressed it in our previous revision.

As previously stated, we are not including children and young people under the age of 18 years in the core outcomes process. This is a considered decision based on our research experience and also the professional and clinical experience of the study advisory team. At the outset of the study, we explored access to families receiving support following child maltreatment with a member of our expert advisory group (community paediatrician and health lead; Lighthouse <https://www.barnahus.eu/en/wp-content/uploads/2020/09/Lighthouse-Annual-Report-2019-web-version.pdf>). It was her clinical opinion that it would be inappropriate to approach families currently receiving support and that she would be unable to facilitate access through her service. Similarly, DVA services within our network were unable to facilitate direct access to children and young people as part of the study. However, CM and DVA organisations were agreeable to advertising the study and the opportunity for participation through their networks of adult survivors. This has resulted in recruitment of a significant number of people with experience of CM and/or DVA as children or more recently as parents of children who have experienced either or both adversity.

It is also worth restating that in phase one of the study (outcome identification) we extracted outcomes and aspirations from child focussed qualitative studies (whether standalone or nested within

intervention studies) and therefore the child's voice is to some degree represented within the longlist of outcomes developed from this study.

As part of our PPI approach, we reached out to several survivor-led groups and frontline support organisations with their own lived experience groups, including those working with children and young people. However, because of the pandemic, most frontline organisations we contacted did not have the time to take part, despite our offer to pay well above current involvement recommendations. We do however have one group of care-experienced young adults involved (aged 18 to 25 years) who we have been consulting throughout the process. We felt that they were able to reflect CYP views, whilst also no longer living in violent and abusive situations, so from a safeguarding perspective this was more straightforward for us as researchers. Our concern, particularly during the pandemic, was that we did not have the structures in place or the resources to ensure the safe participation of CYP under the age of 18 years with recent or current experience of violence and abuse. We have added a paragraph to the participants section in the method, clearly stating the lack of involvement of CYP in the study and providing justification for this. As noted by the reviewer this is also highlighted as a limitation of the study in the discussion section.